

# Observer based robust $H_\infty$ fuzzy tracking control: application to an activated sludge process

Abdelmounaim Khallouq, Asma Karama and Mohamed Abyad

Automation of Environment and Transfer Processes Laboratory/Faculty of Science Semlalia, Cadi Ayyad University, Marrakech, Morocco

## ABSTRACT

The design of an observer-based robust tracking controller is investigated and successfully applied to control an Activated Sludge Process (ASP) in this study. To this end, the Takagi–Sugeno (TS) fuzzy modeling is used to describe the dynamics of a nonlinear system with disturbance. Since the states of the system are not fully available, a fuzzy observer is designed. Based on the observed states and a reference state model, a reduced fuzzy controller for trajectory tracking purposes is then proposed. While the controller and the observer are developed, the design goal is to achieve the convergence and a guaranteed $H_\infty$ performance. By using Lyapunov and $H_\infty$ theories, sufficient conditions for synthesis of a fuzzy observer and a fuzzy controller for TS fuzzy systems are derived. Using some special manipulations, these conditions are reformulated in terms of linear matrix inequalities (LMIs) problem. Finally, the robust and effective tracking performance of the proposed controller is tested through simulations to control the dissolved oxygen and the substrate concentrations in an activated sludge process.

## INTRODUCTION

In the last few years the Takagi-Sugeno (TS) fuzzy modelling, which is a multi-modelling approach, has been emerged as a powerful tool, providing a consistent and efficient approach to handle problems related to modelling and control of nonlinear systems especially wastewater treatment plants (WWTP). These processes are classified as complex systems due to their non-linear dynamics, large uncertainties and the lack of measurements. Hence, it has become a topic of substantial interest exposed to several studies using the TS fuzzy approach. For example, in *Carlos-Hernandez, Beteau & Sanchez (2006)* an application for an anaerobic digestion process has been proposed, where a linearization study involving various representative operating points is first carried out to obtain the TS model then a TS fuzzy observer is designed and experimentally validated. *Nagy Kiss et al. (2011)* proposed a proportional integral observer for uncertain TS fuzzy systems affected by unknown inputs using $L_2$-gain to minimize the effect of the unknown input. The method has been applied on a reduced model of the Activated Sludge Model No. 1 (ASM1). *Belchior, Araújo & Landeck (2012)* proposed the regulation of the dissolved oxygen concentration in WWTP through the implementation of an adaptive

Corresponding author
Abdelmounaim Khallouq,
abdelmounaim.khallouq@gmail.com

fuzzy controller. The study by *Aouaouda et al. (2012)* deals with a fault tolerant control problem of an activated sludge process where an uncertain TS model is considered for the states and faults estimation and used into a robust tracking control scheme using the $L_2$-gain. Recently in *Li et al. (2020)*, the control of the dissolved oxygen is presented using a fuzzy predictive model and where the membership functions of the fuzzy model are obtained based on the fuzzy C-means cluster algorithm.

In parallel, besides stabilization problem, tracking control designs are also important issues for practical applications. There are very successful studies dealing with the output/state tracking control design based on the TS fuzzy approach. In *Lin et al. (2007)* an observer based output tracking control is investigated for TS fuzzy systems with time-delay. *Nachidi, Hajjaji & Bosche (2011)* studied the problem of robust output tracking control of TS fuzzy uncertain discrete-time systems and its application in a DC–DC converters. An adaptive fuzzy control is proposed in *Bououden, Chadli & Karimi (2015)* for uncertain system subject to a pre-treatment of wastewater modeled using the TS approach. A TS fuzzy tracking control problem with respect to input saturation is addressed in *Yu, Lam & Chan (2018)* using an output feedback controller. In *Abyad, Karama & Khallouq (2020)*, an output tracking control problem applied to a fermentation process has been scrutinized by considering the question of asymmetrical constraints on the control inputs. There are also relevant studies for the state tracking control even though its design is more general and more difficult than the design of the output tracking control. For example, in *Senthilkumar & Mahanta (2009)*, a TS fuzzy guaranteed cost controller for trajectory tracking in nonlinear systems is investigated. A fuzzy state feedback law is used to build the controller whose performance is evaluated using a quadratic cost function. By using observers to deal with the absence of full-state information, a robust TS fuzzy observer-based tracking controller is addressed in *Chang & Wu (2012)* where the $H_\infty$ performance is considered to mitigate the tracking error. In our previous study (*Khallouq, Karama & Abyad, 2020*), a robust observer based tracking controller using a reference model is developed where the controller and the observer gains are obtained simultaneously in one single step by solving a set of linear matrix inequalities and where the tracking problem concerns all the state variables.

It is evident that a high number of state variables leads to high order controllers. In fact, a big problem needs to be solved for high order systems which increases numerical computations. Proceeding from the fact that for many realistic applications, it is not necessary to control all state variables. In addition, other difficulties may arise when the system is disturbed the existence of disturbance may deteriorate the performances of the system and can be a source of instability. Therefore, it is more appropriate to develop methodology which involves a low dimensional design for systems with disturbance. Motivated by the discussion above, the presented work is an extension of *Khallouq, Karama & Abyad (2020)*. We propose to design a reduced order observer based state tracking controller in which only the key state variables has been considered and where the

TS fuzzy model has been extended to deal with nonlinear systems with disturbance. Since the system states are not fully accessible to measurement a TS fuzzy observer is used to reconstruct all of them. Because of the nonlinear feature of the bioprocesses dynamics and the usually large uncertainty of some parameters, mainly the kinetic terms and the unknown inputs, the implementation of extended different versions of observers are very promising and have proved to be very successful in several applications e.g., Kalman filter to deal with Gaussian disturbances (*Zeng et al., 2016*; *Silva et al., 2019*) observer based on $H_\infty$ technique (*Katebi, 2001*), the minimum entropy filtering method for non-Gaussian disturbances cases (*Zhang, Chen & Yu, 2017*). The main contribution of this article can be outlined as follows: we propose to split the TS system into two subsystems, one of which involves the part of the state variables to be controlled. Then we reformulate the problem of a robust observer based state tracking control design. The controller is then expressed by a feedback law, which is based on the classical structure of the Parallel Distributed Compensation (PDC) concept, involving the error between the estimate of the controlled state and the state of a reference model. Finally we establish sufficient conditions to guarantee tracking performance for the part to be controlled and stabilization for the remainder part. Unlike *Khallouq, Karama & Abyad (2020)*, here the observer's synthesis is achieved separately from the controller synthesis. Whether for the controller or the observer, the used schemes produce a disturbance term. A performance criterion $H_\infty$ is used in each problem. Based on $H_\infty$ and Lyapunov theory, conditions are developed to ensure convergence and attenuate respectively the tracking error and the estimation error as small as possible. The results are formulated in terms of (*LMIs*). Finally, the efficiency and the robustness of both the tracking control and estimation schemes are demonstrated via simulations on an activated sludge treatment process and are tested under a variety of operating conditions and simulated perturbations.

The paper is organised as follow: First, the modelling of the activated sludge process is described, followed by the design of an Observer-Based Fuzzy Robust Tracking Controller then the design of a robust TS fuzzy observer and end with an application of the proposed method to an activated sludge process.

## The activated sludge process model

The activated sludge schematized in Fig. 1 is used as a biological purification in waste-water treatment, consisting essentially of flocculating microorganisms, mixed with dissolved oxygen and waste-water. Thus, the microorganisms come into contact with the organic pollutants presents in the wastewater, as well as with dissolved oxygen, and are kept in suspension. Based on the natural metabolism, These microorganisms convert the organic matter into new cells, carbon dioxide and water. The process of the purification is done into tow tanks called aerator and settler. The energy required by the process is provided by the dissolved oxygen. Subsequently, carbon dioxide is released in return.

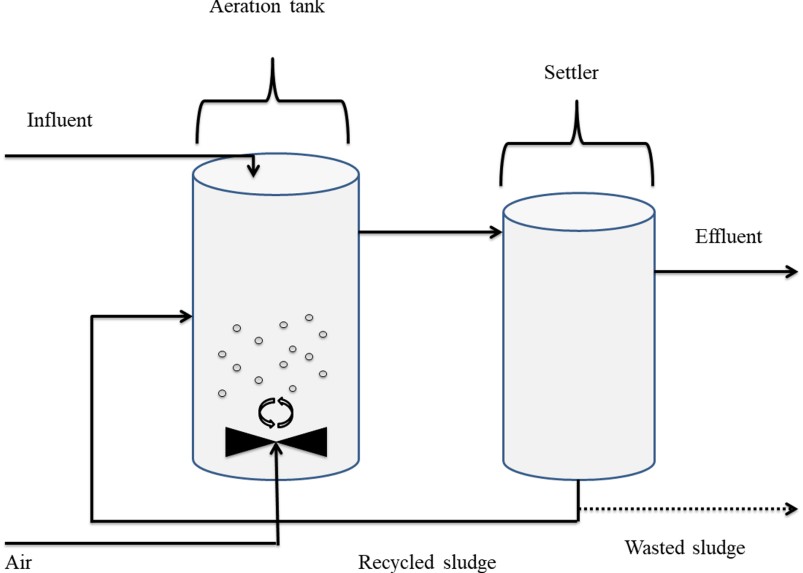

**Figure 1 Schematic diagram of the activated sludge process.**

The mathematical model that represents the process is given using the mass balance around the aerator and the settler as follows (*Nejjari et al., 1999*):

$$
\begin{aligned}
\frac{dX}{dt} &= \mu(.)X - D(1 + q_r)X + q_r D X_r \\
\frac{dS}{dt} &= -\frac{1}{Y}\mu(.)X - (1 + q_r)DS + DS_{in} \\
\frac{dC_o}{dt} &= \frac{K_0}{Y}\mu(.)X - D(1 + q_r)C_o + DC_{oin} + K_{La}(C_s - C_o) \\
\frac{dX_r}{dt} &= D(1 + q_r)X - D(\beta + q_r)X_r
\end{aligned}
\tag{1}
$$

where

- $X(t)$, $S(t)$, $C_o(t)$ and $X_r(t)$ are respectively the biomass, the substrate, the dissolved oxygen and the recycled biomass concentrations.
- $\mu(.)$ corresponds to the biomass specific growth rate. It is assumed to follow the following model:

$$
\mu(S, C_o) = \mu_{max}\frac{S}{K_s + S}\frac{C_o}{K_c + C_o}
$$

$\mu_{max}$ is the maximum specific growth rate, $K_c$ is the saturation constant and $K_s$ is the affinity constant.

- $D$ and $K_{La}$ represent respectively the dilution rate and the aeration flow rate.
- $S_{in}$ and $C_{oin}$ are the influent substrate and the dissolved oxygen concentrations.

- $Y$ is a constant yield coefficient, $K_0$ is a constant and $C_s$ is the maximum concentration of the dissolved oxygen concentration.
- $q_r$ and $\beta$ represent respectively the ratio of recycled flow and the ratio of waste flow to influent flow.

For this model the states, the inputs and the output vectors are given respectively by:

$$x(t) = \begin{bmatrix} X(t) & X_r(t) & S(t) & C_o(t) \end{bmatrix}^T$$
$$u(t) = \begin{bmatrix} D(t) & K_{La}(t) \end{bmatrix}^T \tag{2}$$
$$y(t) = C_o(t)$$

## OBSERVER BASED FUZZY ROBUST TRACKING CONTROLLER DESIGN

### Problem formulation and preliminaries

The TS fuzzy approach consists in transcribing the dynamic of a nonlinear process into a finite weighted sum of linear models. There exist three approaches in the literature to obtain the TS fuzzy model (*Tanaka & Wang, 2003b*): the black box identification, the linearization technique and non-linearity sector method. We are interested in the third method which gives an accurate TS fuzzy model description of nonlinear model without information loss.

Let consider the following nonlinear disturbed system:

$$\dot{x}(t) = f(x(t), u(t), d(t))$$
$$y(t) = Cx(t) \tag{3}$$

where $x(t) \in R^n$ is the state vector, $u(t) \in R^m$ is the input vector, $d(t) \in R^l$ is the disturbance, $y(t) \in R^q$ is the output vector and $C$ a matrix $\in R^{q \times n}$ Consider a TS fuzzy model. The system (3) can be approximated or represented (according to the number $n_r$ of sub-models) by the Takagi–Sugeno structure:

$$\dot{x}(t) = \sum_{i=1}^{n_r} h_i(z)(A_i x(t) + B_i u(t) + Gd(t)) \tag{4}$$

where $A_i \in R^{n \times n}$, $B_i \in R^{n \times m}$ and $G \in R^{n,l}$. $z \in R^p$ denotes the so-called decision variables (premise variables) that can be available when it depends on measurable variable such as $u(t)$ or $y(t)$ i.e., $z = z(u(t), y(t))$ or unavailable when it depends on non-measured system state $x(t)$ i.e., $z = z(x(t))$. The weighting functions $h_i(z)$ called the membership functions satisfy the convex sum property expressed in the following equations:

$$\sum_{i=1}^{n_r} h_i(z) = 1 \quad 0 \le h_i(z) \le 1$$

The weighing functions $h_i(z)$ are generally nonlinear and depend on the premise variables $z$. Let us consider the following partition $x(t) = \begin{bmatrix} x_1(t) \\ x_2(t) \end{bmatrix}$ of the system (4) with

the corresponding matrices $A_i = \begin{bmatrix} A_i^{11} & A_i^{12} \\ A_i^{21} & A_i^{22} \end{bmatrix}$, $B_i = \begin{bmatrix} B_i^1 \\ B_i^2 \end{bmatrix}$, $G = \begin{bmatrix} G^1 \\ G^2 \end{bmatrix}$ and where $x_2(t)$ corresponds to the part of the state vector to be controlled and $x_1(t)$ corresponds to the remaining state variables.

The dynamical model can be rewritten as follows:

$$\dot{x}_1(t) = \sum_{i=1}^{n_r} h_i(z)(A_i^{11}x_1(t) + A_i^{12}x_2(t) + B_i^1 u(t) + G^1 d(t)) \tag{5a}$$

$$\dot{x}_2(t) = \sum_{i=1}^{n_r} h_i(z)(A_i^{21}x_1(t) + A_i^{22}x_2(t) + B_i^2 u(t) + G^2 d(t)) \tag{5b}$$

Consider a linear reference model given by the following equation:

$$\dot{x}_2^r(t) = A_r x_2^r(t) + r(t) \tag{6}$$

where $x_2^r(t)$ is the reference state which should be tracked by the system (5b). $A_r$ is a stable matrix and $r(t)$ is a bounded input reference.

Our goal is to synthesize a control law based on the state estimation capable to reduce the error between the reference trajectory $x_2^r(t)$ and the state $x_2(t)$. The Parallel Distributed Compensation concept can be used to design a fuzzy controller where the main idea consists to design a local controller for each sub-model based on local control rule, which shares with the fuzzy model the same fuzzy sets.

The controller we choose in this paper is expressed by an observer-based law with reference model as follows:

$$u(t) = \sum_{i=1}^{n_r} h_i(\hat{z}) K_i(\hat{x}_2(t) - x_2^r(t)) \tag{7}$$

where $\hat{x}_2(t)$ is the estimation of $x_2(t)$ and the $K_i$'s represent the local feedback gains that should be determined.

## Stability conditions

Before starting the stability analysis, some useful lemmas are recalled.

**Lemma 1** (*Guerra et al., 2006*) *For any matrices X,Y of appropriate dimensions and for any positive scalar* $\eta$ *the following inequality holds:*

$$X^T Y + Y^T X \le \eta X^T X + \eta^{-1} Y^T Y \tag{8}$$

**Lemma 2** (*Guerra et al., 2006*) *Considering* $\pi$ *(as in inequality (9))* $< 0$ *a matrix X and a scalar* $\lambda$, *the following inequality holds:*

$$X^T \Pi X \le -\lambda(X^T + X) - \lambda^2 \Pi^{-1} \tag{9}$$

Using (7) into (5), the closed loop system can then be rewritten as follows:

$$\dot{x}_1(t) = \sum_{i,j=1}^{n_r} h_i(\hat{z})h_j(\hat{z})(A_i^{11}x_1(t) + A_i^{12}x_2(t) + B_i^1 K_j(\hat{x}_2(t) - x_2^r(t))) + \omega_1(t)$$

$$\dot{x}_2(t) = \sum_{i,j=1}^{n_r} h_i(\hat{z})h_j(\hat{z})(A_i^{21}x_1(t) + A_i^{22}x_2(t) + B_i^2 K_j(\hat{x}_2(t) - x_2^r(t))) + \omega_2(t)$$

(10)

where

$$\omega_1(t) = \sum_{i=1}^{n_r}(h_i(z) - h_i(\hat{z}))(A_i^{11}x_1(t) + A_i^{12}x_2(t) + B_i^1 u(t) + G^1 d(t))$$

$$\omega_2(t) = \sum_{i=1}^{n_r}(h_i(z) - h_i(\hat{z}))(A_i^{21}x_1(t) + A_i^{22}x_2(t) + B_i^2 u(t) + G^2 d(t))$$

Let us define by $e_r(t) = x_2(t) - x_2^r(t)$ the tracking error and $e_{o2}(t) = x_2(t) - \hat{x}_2(t)$ the state estimation error and consider the augmented state: $x_a(t) = \begin{bmatrix} x_1(t) \\ e_r(t) \end{bmatrix}$. Using (10), the dynamical model of the augmented system is given by:

$$\dot{x}_a(t) = \sum_{i=1}^{n_r}\sum_{j=1}^{n_r} h_i(\hat{z})h_j(\hat{z})(\mathscr{A}_{ij}x_a(t) + \mathscr{D}_{ij}\xi(t))$$

(11)

which can be rewritten as:

$$\dot{x}_a(t) = \sum_{i=1}^{n_r} h_i^2(\hat{z})(\mathscr{A}_{ii}x_a(t) + \mathscr{D}_{ii}\xi(t)) + 2\sum_{\substack{i,j=1 \\ i<j}}^{n_r} h_i(\hat{z})h_j(\hat{z})\left(\frac{\mathbb{A}_{ij}x_a(t) + \mathbb{D}_{ij}\xi(t)}{2}\right)$$

(12)

where

$$\mathscr{A}_{ij} = \begin{bmatrix} A_i^{11} & A_i^{12} + B_i^1 K_j \\ A_i^{21} & A_i^{22} + B_i^2 K_j \end{bmatrix}, \quad \mathbb{A}_{ij} = \mathscr{A}_{ij} + \mathscr{A}_{ji}$$

$$\mathscr{D}_{ij} = \begin{bmatrix} -B_i^1 K_j & A_i^{12} & 0 & I & 0 \\ -B_i^2 K_j & (A_i^{22} - A_{ref}) & -I & 0 & I \end{bmatrix}, \quad \mathbb{D}_{ij} = \mathscr{D}_{ij} + \mathscr{D}_{ji}$$

(13)

$$\xi(t) = \begin{bmatrix} e_{o2}(t)^T & x_2^r(t)^T & r(t)^T & \omega_1(t)^T & \omega_2(t)^T \end{bmatrix}^T$$

The term $\xi(t)$ is acting like a disturbance affecting the augmented state $x_a(t)$. Thus to attenuate its effect, we propose the use of the $H_\infty$ technique applied to System (12). The weighted $H_\infty$ performance to minimize can be presented as follows:

$$\int_0^{t_f} x_a^T(t)Qx_a(t)dt \leq \gamma^2 \int_0^{t_f} \xi^T(t)\xi(t)dt$$

(14)

where $Q$ is a positive definite matrix and $\gamma$ is a prescribed positive scalar that defines the attenuation level of the disturbance $\xi(t)$.

To realize Condition (14), one has to use a Lyapunov function for System (12) given by:

$$V(x_a) = x_a(t)^T P x_a(t) \quad (P > 0) \tag{15}$$

To achieve the performance (14) and ensure the stability of augmented System (12), the following condition must be realized:

$$\dot{V}(x_a) + x_a^T(t) Q x_a(t) - \gamma^2 \xi(t)^T \xi(t) \leq 0 \tag{16}$$

The following result can be announced:

**Lemma 3** *If there exist positive definite matrices P, $Q_1$ and $Q_2$, and positive scalars $\alpha_1, \alpha_2$, $\gamma_1$ and $\gamma_2$, the augmented system in* (12) *is asymptotically stable, such that the following conditions are satisfied:*

$$\begin{aligned}
\mathscr{A}_{ii}^T P + P\mathscr{A}_{ii} + Q_1 + \alpha_1^{-1} P\mathscr{D}_{ii}\mathscr{D}_{ii}^T P &< 0 \quad \forall i = 1, \ldots, n_r \\
\alpha_1 - \gamma_1 &< 0
\end{aligned} \tag{17}$$

$$\begin{aligned}
\tfrac{1}{2}\left(\mathbb{A}_{ij}^T P + P\mathbb{A}_{ij} + Q_2 + \alpha_2^{-1} P\mathbb{D}_{ij}\mathbb{D}_{ij}^T P\right) &< 0 \quad \forall i < j = 1, \ldots, n_r \\
\tfrac{1}{2}(\alpha_2 - \gamma_2) &< 0
\end{aligned} \tag{18}$$

The $H_\infty$ performmance criteria (14) is guaranteed where the scalar $\gamma$ and the matrix $Q$ are given by $\gamma = \sqrt{\gamma_1 + \gamma_2}$ and $Q = Q_1 + Q_2$

**Proof:**

Using (12), the derivative of the Lyapunov function (16) is:

$$\begin{aligned}
\dot{V}(x_a) = &\sum_{i=1}^{n_r} h_i^2(\hat{z})[x_a^T(t)(\mathscr{A}_{ii}^T P + P\mathscr{A}_{ii})x_a(t) + \xi^T(t)\mathscr{D}_{ii}^T P x_a(t) + x_a^T(t)P\mathscr{D}_{ii}\xi(t)] + \\
&2 \sum_{\substack{i,j=1 \\ i<j}}^{n_r} h_i(\hat{z})h_j(\hat{z})\frac{1}{2}[x_a^T(t)(\mathbb{A}_{ij}^T P + P\mathbb{A}_{ij})x_a(t) + \xi^T(t)\mathbb{D}_{ij}^T P x_a(t) + x_a^T(t)P\mathbb{D}_{ij}\xi(t)]
\end{aligned} \tag{19}$$

Denote by J the expression:

$$J = \dot{V}(x_a(t)) + x_a^T(t) Q x_a(t) - \gamma^2 \xi(t)^T \xi(t) \tag{20}$$

by setting $Q = Q_1 + Q_2$ and $\gamma^2 = \gamma_1 + \gamma_2$ and using (19), J can be written as the sum of two terms:

$$\begin{aligned}
J = &\sum_{i=1}^{n_r} h_i^2(\hat{z})[x_a^T(t)(\mathscr{A}_{ii}^T P + P\mathscr{A}_{ii})x_a(t) + x_a^T(t)Q_1 x_a(t) - \gamma_1 \xi^T(t)\xi(t)) \\
&+ \xi^T(t)\mathscr{D}_{ii}^T P x_a(t) + x_a^T(t)P\mathscr{D}_{ii}\xi(t)] \\
&+ 2 \sum_{\substack{i,j=1 \\ i<j}}^{n_r} h_i(\hat{z})h_j(\hat{z})\frac{1}{2}[x_a^T(t)(\mathbb{A}_{ij}^T P + P\mathbb{A}_{ij})x_a(t) + x_a^T(t)Q_2 x_a(t) - \gamma_2 \xi^T(t)\xi(t) \\
&+ \xi^T(t)\mathbb{D}_{ij}^T P x_a(t) + x_a^T(t)P\mathbb{D}_{ij}\xi(t)]
\end{aligned} \tag{21}$$

Using the Lemma 1 on the crossed terms yields:

$$\begin{aligned}
\xi^T(t)\mathscr{D}_{ii}^T P x_a(t) + x_a^T(t) P \mathscr{D}_{ii}\xi(t) &\leq \alpha_1^{-1} x_a^T(t) P \mathscr{D}_{ii}\mathscr{D}_{ii}^T P x_a(t) + \alpha_1 \xi^T(t)\xi(t) \\
\xi^T(t)\mathbb{D}_{ij}^T P x_a(t) + x_a^T(t) P \mathbb{D}_{ij}\xi(t) &\leq \alpha_2^{-1} x_a^T(t) P \mathbb{D}_{ij}\mathbb{D}_{ij}^T P x_a(t) + \alpha_2 \xi^T(t)\xi(t)
\end{aligned} \tag{22}$$

(22) into (21) leads to the following inequality:

$$\begin{aligned}
J \leq \;& \sum_{i=1}^{n_r} h_i^2(\hat{z})[x_a^T(t)(\mathscr{A}_{ii}^T P + P\mathscr{A}_{ii} + Q_1 + \alpha_1^{-1}P\mathscr{D}_{ii}\mathscr{D}_{ii}^T P)x_a(t) + \xi^T(t)(\alpha_1 - \gamma_1)\xi(t))] + \\
& 2\sum_{\substack{i,j=1 \\ i<j}}^{n_r} h_i(\hat{z})h_j(\hat{z})\frac{1}{2}[x_a^T(t)(\mathbb{A}_{ij}^T P + P\mathbb{A}_{ij} + Q_2 + \alpha_2^{-1}P\mathbb{D}_{ij}\mathbb{D}_{ij}^T P)x_a(t) \\
& + \xi^T(t)(\alpha_2 - \gamma_2)\xi(t)]
\end{aligned} \tag{23}$$

This implies that (16) is satisfied if the following sufficient conditions hold:

$$\begin{aligned}
\mathscr{A}_{ii}^T P + P\mathscr{A}_{ii} + Q_1 + \alpha_1^{-1}P\mathscr{D}_{ii}\mathscr{D}_{ii}^T P &< 0 \quad \forall i = 1, \dots, n_r \\
\alpha_1 - \gamma_1 &< 0
\end{aligned} \tag{24}$$

$$\begin{aligned}
\tfrac{1}{2}(\mathbb{A}_{ij}^T P + P\mathbb{A}_{ij} + Q_2 + \alpha_3^{-1}P\mathbb{D}_{ij}\mathbb{D}_{ij}^T P) &< 0 \quad \forall i < j = 1, \dots, n_r \\
\tfrac{1}{2}(\alpha_2 - \gamma_2) &< 0
\end{aligned} \tag{25}$$

This ends the lemma proof.

## The main result

To determine the controller gains $K_i$, we present new conditions in terms of LMIs. These conditions are developed through the use of separation Lemma 2, the introduction of some slack variables and other calculations leading to the following results.

**Theorem 1** *There exists an observer based controller (7) for the system (12) guaranteeing the $H_\infty$ performance criteria (14) if there exists positive matrices $X_1 = X_1^T, X_2 = X_2^T$, matrices $Y_i, i = 1,2,\dots,n_r$, positive matrices $\tilde{Q}_1^1, \tilde{Q}_1^2, \tilde{Q}_2^1$ and $\tilde{Q}_2^2$ and prescribed positive scalars $\alpha_1, \alpha_2, \alpha_3, \alpha_4, \gamma_1$ and $\gamma_2$ such that $\alpha_1 < \gamma_1$ and $\alpha_2 < \gamma_2$ and that the following conditions hold: for $i = 1,\dots,n_r$*

$$\begin{bmatrix}
M_1 & M_2 & -B_i^1 Y_i & A_i^{12} & 0 & I & 0 & 0 \\
* & M_3 & -B_i^2 Y_i & A_i^{22} - A_r & -I & 0 & I & 0 \\
* & * & -2\alpha_3 X_2 & 0 & 0 & 0 & 0 & -\alpha_3 I \\
* & * & * & -\alpha_1 I & 0 & 0 & 0 & 0 \\
* & * & * & * & -\alpha_1 I & 0 & 0 & 0 \\
* & * & * & * & * & -\alpha_1 I & 0 & 0 \\
* & * & * & * & * & * & -\alpha_1 I & 0 \\
* & * & * & * & * & * & * & -\alpha_1 I
\end{bmatrix} < 0 \tag{26}$$

for $i < j = 1, \dots, n_r$

$$\frac{1}{2}\begin{bmatrix} N_1 & N_2 & -B_i^1 Y_j - B_j^1 Y_i & A_i^{12} + A_j^{12} & 0 & 2I & 0 & 0 \\ * & N_3 & -B_i^2 Y_j - B_j^2 Y_i & A_i^{22} + A_j^{22} - 2A_r & -2I & 0 & 2I & 0 \\ * & * & -2\alpha_4 X_2 & 0 & 0 & 0 & 0 & -\alpha_4 I \\ * & * & * & -\alpha_2 I & 0 & 0 & 0 & 0 \\ * & * & * & * & -\alpha_2 I & 0 & 0 & 0 \\ * & * & * & * & * & -\alpha_2 I & 0 & 0 \\ * & * & * & * & * & * & -\alpha_2 I & 0 \\ * & * & * & * & * & * & * & -\alpha_2 I \end{bmatrix} < 0 \quad (27)$$

where

$$M_1 = A_i^{11} X_1 + X_1 A_i^{11^T} + \tilde{Q}_1^1$$

$$M_2 = X_1 A_i^{21^T} + A_i^{12} X_2 + B_i^1 Y_i$$

$$M_3 = A_i^{22} X_2 + X_2 A_i^{22^T} + B_i^2 Y_i + (B_i^2 Y_i)^T + \tilde{Q}_1^2$$

$$N_1 = (A_i^{11} + A_j^{11}) X_1 + X_1 (A_i^{11} + A_j^{11})^T + \tilde{Q}_2^1$$

$$N_2 = X_1 (A_i^{21} + A_j^{21})^T + (A_i^{12} + A_j^{12}) X_2 + B_i^1 Y_j + B_j^1 Y_i$$

$$N_3 = (A_i^{22} + A_j^{22}) X_2 + X_2 (A_i^{22} + A_j^{22})^T + (B_i^2 Y_j + B_j^2 Y_i) + \tilde{Q}_2^2 + (B_i^2 Y_j + B_j^2 Y_i)^T$$

*and* * *stands for the symmetric term of the corresponding off-diagonal term.*

Solving LMIs (26) and (27) the controller gains $K_i$, the attenuation level $\gamma$ and the matrix $Q$ are given by:$K_i = Y_i X_2^{-1}, i = 1, \ldots, n_r$ and $\gamma = \sqrt{\gamma_1 + \gamma_2}$ $Q = Q_1 + Q_2$ where

$$Q_1 = \begin{bmatrix} Q_1^1 & 0 \\ 0 & Q_1^2 \end{bmatrix} \text{ and } Q_2 = \begin{bmatrix} Q_2^1 & 0 \\ 0 & Q_2^2 \end{bmatrix}$$

**Proof:**

For the proof of the theorem 1, we will start from the sufficient conditions given in the Lemma 3. Let us consider the first condition (17). Multiplying it post and prior by $P^{-1}$ and using Schur lemma, the following inequality is obtained:

$$\begin{bmatrix} P^{-1}\mathscr{A}_{ii}^T + \mathscr{A}_{ii} P^{-1} + \tilde{Q}_1 & \mathscr{D}_{ii} \\ \mathscr{D}_{ii}^T & -\alpha_1 I \end{bmatrix} < 0 \quad (28)$$

By choosing matrices $P$ and $Q_1$ as follows: $P = \begin{bmatrix} P_1 & 0 \\ 0 & P_2 \end{bmatrix}$ and $\tilde{Q}_1 = P^{-1} Q_1 P^{-1} = \begin{bmatrix} \tilde{Q}_1^1 & 0 \\ 0 & \tilde{Q}_1^2 \end{bmatrix}$ and Replacing $\mathscr{D}_{ii}, \mathscr{A}_{ii}$ by their expressions in (13), (28) becomes:

$$\begin{bmatrix} M_1 & M_2 & -B_i^1 K_i & A_i^{12} & 0 & I & 0 \\ * & M_3 & -B_i^2 K_i & A_i^{22} - A_r & -I & 0 & I \\ * & * & -\alpha_1 I & 0 & 0 & 0 & 0 \\ * & * & * & -\alpha_1 I & 0 & 0 & 0 \\ * & * & * & * & -\alpha_1 I & 0 & 0 \\ * & * & * & * & * & -\alpha_1 I & 0 \\ * & * & * & * & * & * & -\alpha_1 I \end{bmatrix} < 0 \quad (29)$$

where

$$M_1 = A_i^{11} P_1^{-1} + P_1^{-1} A_i^{11^T} + \tilde{Q}_1^1$$

$$M_2 = P_1^{-1} A_i^{21^T} + (A_i^{12} + B_i^1 K_i) P_2^{-1}$$

$$M_3 = (A_i^{22} + B_i^2 K_i) P_2^{-1} + P_2^{-1} (A_i^{22} + B_i^2 K_i)^T + \tilde{Q}_1^2$$

Multiplying (29) left and right respectively by $diag(\begin{bmatrix} I & I & P_2^{-1} & I & I & I & I \end{bmatrix})$ and its transpose yields to:

$$\begin{bmatrix} M_1 & M_2 & -B_i^1 K_i P_2^{-1} & A_i^{12} & 0 & I & 0 \\ * & M_3 & -B_i^2 K_i P_2^{-1} & A_i^{22} - A_r & -I & 0 & I \\ * & * & -P_2^{-1} \alpha_1 P_2^{-1} & 0 & 0 & 0 & 0 \\ * & * & * & -\alpha_1 I & 0 & 0 & 0 \\ * & * & * & * & -\alpha_1 I & 0 & 0 \\ * & * & * & * & * & -\alpha_1 I & 0 \\ * & * & * & * & * & * & -\alpha_1 I \end{bmatrix} < 0 \tag{30}$$

Using Lemma 2 we have:

$$P_2^{-1}(-\alpha_1) P_2^{-1} \leq -2\alpha_3 P_2^{-1} + \alpha_3^2 (\alpha_1)^{-1} I \tag{31}$$

and Schur complement yields to:

$$\begin{bmatrix} M_1 & M_2 & -B_i^1 K_i P_2^{-1} & A_i^{12} & 0 & I & 0 & 0 \\ * & M_3 & -B_i^2 K_i P_2^{-1} & A_i^{22} - A_r & -I & 0 & I & 0 \\ * & * & -\alpha_3 P_2^{-1} & 0 & 0 & 0 & 0 & \alpha_3 I \\ * & * & * & -\alpha_1 I & 0 & 0 & 0 & 0 \\ * & * & * & * & -\alpha_1 I & 0 & 0 & 0 \\ * & * & * & * & * & -\alpha_1 I & 0 & 0 \\ * & * & * & * & * & * & -\alpha_1 I & 0 \\ * & * & * & * & * & * & * & -\alpha_1 \end{bmatrix} < 0 \tag{32}$$

Using the following variable change $X_1 = P_1^{-1}, X_2 = P_2^{-1}$ and $Y_i = K_i X_2$, the conditions (26) of the theorem is fulfilled.

To carry out the second LMI of the theorem 1, we proceed in the same way with the second sufficient condition of Lemma 3. Multiplying (18) post and prior with $P^{-1}$ and using the Schur lemma, the following inequality is obtained:

$$\frac{1}{2} \begin{bmatrix} P^{-1} \mathbb{A}_{ij}^T + \mathbb{A}_{ij} P^{-1} + \tilde{Q}_2 & \mathbb{D}_{ij} \\ \mathbb{D}_{ij}^T & -\alpha_2 I \end{bmatrix} < 0 \tag{33}$$

By choosing the matrix $\tilde{Q}_2 = P^{-1}Q_2P^{-1} = \begin{bmatrix} \tilde{Q}_2^1 & 0 \\ 0 & \tilde{Q}_2^2 \end{bmatrix}$ and replacing $\mathbb{D}_{ij}$ and $\mathbb{A}_{ij}$ by their expression in (13), (33) becomes:

$$\frac{1}{2}\begin{bmatrix} N_1 & N_2 & -(B_i^1 K_j + B_j^1 K_i) & A_i^{12} + A_j^{12} & 0 & 2I & 0 \\ * & N_3 & -(B_i^2 K_j + B_j^2 K_i) & A_i^{22} + A_j^{22} - 2A_r & -2I & 0 & 2I \\ * & * & -\alpha_2 I & 0 & 0 & 0 & 0 \\ * & * & * & -\alpha_2 I & 0 & 0 & 0 \\ * & * & * & * & -\alpha_2 I & 0 & 0 \\ * & * & * & * & * & -\alpha_2 I & 0 \\ * & * & * & * & * & * & -\alpha_2 I \end{bmatrix} < 0 \qquad (34)$$

$$N_1 = (A_i^{11} + A_j^{11})P_1^{-1} + P_1^{-1}(A_i^{11} + A_j^{11})^T + \tilde{Q}_2^1$$

$$N_2 = P_1^{-1}(A_i^{21} + A_j^{21})^T + (A_i^{12} + A_j^{12})P_2^{-1} + (B_i^1 K_j + B_j^1 K_i)P_2^{-1}$$

$$N_3 = (A_i^{22} + A_j^{22} + B_i^2 K_j + B_j^2 K_i)P_2^{-1} + P_2^{-1}(A_i^{22} + A_j^{22} + B_i^2 K_j + B_j^2 K_i)^T + \tilde{Q}_2^2$$

Multiplying (34) left and right respectively by: $diag(\begin{bmatrix} I & I & P_2^{-1} & I & I & I & I \end{bmatrix})$ and its transpose we get:

$$\frac{1}{2}\begin{bmatrix} N_1 & N_2 & -(B_i^1 K_j + B_j^1 K_i)P_2^{-1} & A_i^{12} + A_j^{12} & 0 & 2I & 0 \\ * & N_3 & -(B_i^2 K_j + B_j^2 K_i)P_2^{-1} & A_i^{22} + A_j^{22} - 2A_r & -2I & 0 & 2I \\ * & * & -P_2^{-1}\alpha_2 P_2^{-1} & 0 & 0 & 0 & 0 \\ * & * & * & -\alpha_2 I & 0 & 0 & 0 \\ * & * & * & * & -\alpha_2 I & 0 & 0 \\ * & * & * & * & * & -\alpha_2 I & 0 \\ * & * & * & * & * & * & -\alpha_2 I \end{bmatrix} < 0 \quad (35)$$

using Lemma 2 we have:

$$P_2^{-1}(-\alpha_2)P_2^{-1} \leq -2\alpha_4 P_2^{-1} + \alpha_4^2(\alpha_2)^{-1}I \qquad (36)$$

and Schur complements yiels to:

$$\frac{1}{2}\begin{bmatrix} N_1 & N_2 & -(B_i^1 K_j + B_j^1 K_i)P_2^{-1} & A_i^{12} + A_j^{12} & 0 & 2I & 0 & 0 \\ * & N_3 & -(B_i^2 K_j + B_j^2 K_i)P_2^{-1} & A_i^{22} + A_j^{22} - 2A_r & -2I & 0 & 2I & 0 \\ * & * & -2\alpha_4 P_2^{-1} & 0 & 0 & 0 & 0 & \alpha_4 I \\ * & * & * & -\alpha_2 I & 0 & 0 & 0 & 0 \\ * & * & * & * & -\alpha_2 I & 0 & 0 & 0 \\ * & * & * & * & * & -\alpha_2 I & 0 & 0 \\ * & * & * & * & * & * & -\alpha_2 I & 0 \\ * & * & * & * & * & * & * & -\alpha_2 I \end{bmatrix} < 0 \quad (37)$$

Using the following variable change $X_1 = P_1^{-1}, X_2 = P_2^{-1}$ and $Y_i = K_i X_2$, the conditions (27) of the theorem is fulfilled. This achieves the proof of the theorem.

### A Robust TS fuzzy observer design

In order to estimate both the state variables and the disturbance the following augmented state vector is considered $\bar{x}(t) = \begin{bmatrix} x^T(t) & d^T(t) \end{bmatrix}^T$. From the TS system (4) we have:

$$\dot{\bar{x}}(t) = \sum_{i=1}^{n_r} h_i(\hat{z}) \left( \begin{bmatrix} A_i & G \\ 0 & 0 \end{bmatrix} \bar{x}(t) + \begin{bmatrix} B_i \\ 0 \end{bmatrix} u(t) + \begin{bmatrix} \omega(t) \\ \dot{d}(t) \end{bmatrix} \right)$$
$$= \sum_{i=1}^{n_r} h_i(\hat{z})(\bar{A}_i \bar{x}(t) + \bar{B}_i u(t) + \bar{\omega}(t)) \tag{38}$$
$$y(t) = \bar{C} \bar{x}(t)$$

With:

$$\bar{A}_i = \begin{bmatrix} A_i & G \\ 0 & 0 \end{bmatrix}, \quad \bar{B}_i = \begin{bmatrix} B_i \\ 0 \end{bmatrix}, \quad \bar{C} = \begin{bmatrix} C & 0 \end{bmatrix}, \quad \bar{\omega}(t) = \begin{bmatrix} \omega(t) \\ \dot{d}(t) \end{bmatrix}$$

and

$$\omega(t) = \sum_{i=1}^{n_r} (h_i(z) - h_i(\hat{z}))(A_i x(t) + B_i u(t) + G d(t))$$

Let consider the following fuzzy Luenberger observer (*Tanaka & Wang, 2003a*) for the system (38):

$$\dot{\hat{\bar{x}}}(t) = \sum_{i=1}^{n_r} h_i(\hat{z})(\bar{A}_i \hat{\bar{x}}(t) + \bar{B}_i u(t) + \bar{L}_i \bar{C}(\bar{x}(t) - \hat{\bar{x}}(t))) \tag{39}$$

where the $\bar{L}'_i s$ represent the local observer gains. Note that this observer considers that the premise variables are unknown.

The dynamical model of the estimation error is then given by:

$$\dot{e}_o(t) = \dot{\bar{x}}(t) - \dot{\hat{\bar{x}}}(t) = \sum_{i=1}^{n_r} h_i(\hat{z})(\bar{A}_i - \bar{L}_i \bar{C}) \bar{e}_o(t) + \bar{\omega}(t) \tag{40}$$

where $\bar{\omega}(t)$ is acting as disturbance. To attenuate its effect, the following $H_\infty$ performance is used:

$$\int_0^{t_f} \bar{e}_o^T(t) R \bar{e}_o(t) dt \leq v^2 \int_0^{t_f} \bar{\omega}(t)^T \bar{\omega}(t) dt \tag{41}$$

where $R$ is a positive definite matrix and $v$ is the attenuation level of the disturbances $\bar{\omega}(t)$.

**Theorem 2** *Prescribing the attenuation level v, an observer (39) for the system (46) satisfying the $H_\infty$ performance criterion (41) exist if there exists a symmetric and positive matrix $P_o = P_o^T > 0$, a matrix $R > 0$ and $Z_i$, i = 1,2,…,r and a positive scalar $\eta$ such that the following LMIs are feasible*

$$\begin{bmatrix} P_o \bar{A}_i - Z_i \bar{C} + (P_o \bar{A}_i - Z_i \bar{C})^T + R & P_o \\ P_o & -\eta I \end{bmatrix} < 0 \tag{42}$$

$$\eta - v^2 < 0 \tag{43}$$

Solving LMIs 42, the observer gains $\bar{L}_i$ are given by: $\bar{L}_i = P_o^{-1} Z_i, i = 1, \ldots, n_r$. The scalar verifying the $H_\infty$ norm for the observer is given by: $v = \sqrt{v^2}$

**Proof:** See Appendix A.

## Application to an Activated Sludge Process

This article addresses the problem of controlling an activated sludge treatment process using the TS approach developed above. The dissolved oxygen concentration in the ASP is an important parameter in the process control that has a considerable effect on the treatment effectiveness and economical cost. The reduction of organic substrate concentration is also crucially important and presents one of the main issues in the treatment process. It helps keeping a hight effluent quality. The control objective is to force the substrate and the dissolved oxygen concentrations $S(t)$ and $C_o(t)$ to track the states of a given reference model $S^{ref}(t)$ and $C_o^{ref}(t)$ under the following conditions:

- The dilution rate $D(t)$ and the aeration flow rate $K_{La}(t)$ are the control variables.
- The dissolved oxygen concentration $C_o(t)$ is available.
- Biomass, substrate and recycled biomass concentrations $X(t)$, $S(t)$ and $X_r(t)$ are not available online.

## The TS fuzzy model design

To simulate more realistic conditions, the model (1) is rewritten in the form of a disturbed system. It is assumed that the concentration of the influent substrate is varying during a day instead of considering a constant value. The variation $\delta \, Sin(t)$ around the daily average $Sin$ acts as a disturbance.

To build the TS model, the classical transformation of the non-linearity sector method is used. The following non linearities (the premise variables) are considered:

$$
\begin{aligned}
z_1(x) &= z_1(S, C_o) = \mu_{max} \frac{C_o S}{(K_s + S)(K_c + C_o)} - S \\
z_2(x) &= z_2(X) = X \\
z_3(x) &= z_3(X_r) = X_r \\
z_4(x) &= z_4(C_o) = C_o \\
z_5(x) &= z_5(X, S, C_o) = \mu_{max} \frac{C_o X}{(K_s + S)(K_c + C_o)} \\
z_6(x) &= z_6(S) = S
\end{aligned} \tag{44}
$$

The nonlinear model of the Activated Sludge process described by Eq. (1) can be written in the form:

$$
\begin{aligned}
\dot{x}(t) &= A(z_1, z_2, z_3, z_4, z_5)x(t) + B(z_2, z_3, z_4, z_6)u(t) + Gd(t) \\
y(t) &= Cx(t)
\end{aligned} \tag{45}
$$

where the state, input and output vectors are

$$x(t) = \begin{bmatrix} X \\ X_r \\ S \\ C_o \end{bmatrix}, \quad u(t) = \begin{bmatrix} D \\ K_{la} \end{bmatrix}, \quad y = [C_o], \quad G = \begin{bmatrix} 0 \\ 0 \\ 1 \\ 0 \end{bmatrix}, \quad \text{and} \quad d(t) = D\Delta \mathrm{Sin}(t)$$

and the obtained matrices have the general form:

$$A(z_1, z_2, z_3, z_4, z_5) = \begin{bmatrix} z_1 & 0 & z_2 & 0 \\ 0 & -z_4 & 0 & z_3 \\ 0 & 0 & 0 & -\frac{1}{Y}z_5 \\ -z_3 & z_2 & 0 & -\frac{K_0}{Y}z_5 \end{bmatrix};$$

$$B(z_3, z_4, z_6) = \begin{bmatrix} -(1+q_r)z_2 + q_r z_3 & 0 \\ (1+q_r)z_2 - (\beta+q_r)z_3 & 0 \\ S_{in} - (1+q_r)z_6 & 0 \\ C_{oin} - (1+q_r)z_4 & C_s - z_4 \end{bmatrix} \quad \text{and} \quad C = \begin{bmatrix} 0 & 0 & 0 & 1 \end{bmatrix}$$

6 premise variables are considered. Therefore, the model (45) can be represented by $n_r = 2^6 = 64$ TS submodels, which is rewritten as follows:

$$\dot{x}(t) = \sum_{i=1}^{n_r} h_i(z)(A_i x(t) + B_i u(t) + G d(t)) \tag{46}$$

The $A_i$'s and $B_i$'s corresponding matrices and the $h_i(z)$'s membership functions are obtained from $A(z_1, z_2, z_3, z_4, z_5)$ and $B(z_2, z_3, z_4, z_6)$ and (44), for more explanation on the method to obtain them the reader can refer to (*Nagy et al., 2010*)

### Observer based controller synthesis and simulation results

The following partition $x(t) = \begin{bmatrix} x_1(t) \\ x_2(t) \end{bmatrix}$ are chosen with the corresponding vectors:

$x_1(t) = \begin{bmatrix} X \\ X_r \end{bmatrix}$, and $x_2(t) = \begin{bmatrix} S \\ C_o \end{bmatrix}$. For simulation, the matrix $Ar = \begin{bmatrix} -\frac{1}{10} & 0 \\ 0 & -\frac{1}{2} \end{bmatrix}$ is used

to generate the trajectories of the reference state $x_2^r(t) = \begin{bmatrix} S^{ref}(t) \\ C_o^{ref}(t) \end{bmatrix}$ and the simulation

parameters of Table 1 are considered. To test the robustness of the proposed method, a sinusoidal variation of the influent substrate and changes in the kinetic parameters will be introduced during the simulation.

**Remark 1** The LMIs in the theorem 1 and 2 are solved using matlab with the YALMIP toolbox can be downloaded from https://yalmip.github.io/.

• The resolution of the LMIs in theorem 1 for the parameters $\alpha_1 = 0.4$, $\alpha_2 = 0.4$, $\alpha_3 = 300$, $\alpha_4 = 295$, $\gamma_1 = 0.4$ and $\gamma_2 = 0.4$ leads to:

$$X_1 = 10^3 \times \begin{bmatrix} 29435 & 27668 \\ 27668 & 27609 \end{bmatrix}, X_2 = 10^3 \times \begin{bmatrix} 45062 & 3233 \\ 3233 & 21729 \end{bmatrix}$$

**Table 1 Simulation parameters (*Nejjari et al., 1999*).**

| $Y$ | $q_r$ | $\beta$ | $K_0$ | $C_s$ | $S_{in}$ | $C_{oin}$ | $\mu_{max}$ | $K_s$ | $K_c$ |
|-----|-------|---------|-------|-------|----------|-----------|-------------|-------|-------|
| 0.65 | 0.6 | 0.2 | 0.5 | 10 | 200 | 0.5 | 0.15 | 100 | 2 |
|  |  |  |  | [mg/l] | [mg/l] | [mg/l] | $[h^{-1}]$ | [mg/l] | [mg/l] |

$$\tilde{Q}_1^1 = \tilde{Q}_1^2 = \tilde{Q}_2^1 = \tilde{Q}_2^2 = \begin{bmatrix} 1 & 0 \\ 0 & 1 \end{bmatrix}$$

The attenuation level is given by: $\gamma = \sqrt{\gamma_1 + \gamma_2} = 0.8944$, bellow are given some of the controller gains:

$$K_1 = \begin{bmatrix} 0.1957 & -0.1822 \\ 3.1512 & -80.1625 \end{bmatrix}, K_8 = \begin{bmatrix} -0.4025 & -0.0942 \\ 2.4783 & -38.7399 \end{bmatrix}, K_{16} = \begin{bmatrix} 0.0442 & -0.4936 \\ -13.2269 & -23.2598 \end{bmatrix}$$

$$K_{32} = \begin{bmatrix} -0.2180 & 0.0290 \\ 0.3513 & -27.3775 \end{bmatrix}, K_{64} = \begin{bmatrix} -0.1524 & -0.0023 \\ 1.8694 & -22.9140 \end{bmatrix}$$

- The resolution of the LMIs in in theorem 2 for an attenuation level $v = 1.0681$ and $\eta = 1.1408$ leads to:

$$P_o = \begin{bmatrix} 0.0006 & 0.0001 & -0.0027 & 0.0008 & -0.0003 \\ 0.0001 & 0.1353 & -0.0005 & -0.0003 & 0.0039 \\ -0.0027 & -0.0005 & 0.1678 & 0.0004 & -0.1937 \\ 0.0008 & -0.0003 & 0.0004 & 0.0926 & 0.0000 \\ -0.0003 & 0.0039 & -0.1937 & 0.0000 & 0.2851 \end{bmatrix}$$

$$R = \begin{bmatrix} 0.0385 & -0.0278 & -0.0056 & 0.0000 & 0.0006 \\ -0.0278 & 0.0379 & -0.0066 & 0.0000 & -0.0003 \\ -0.0056 & -0.0066 & 0.0598 & 0.0000 & -0.0220 \\ 0.0000 & 0.0000 & 0.0000 & 1.0000 & 0.0000 \\ 0.0006 & -0.0003 & -0.0220 & 0.0000 & 0.0516 \end{bmatrix}$$

- bellow are given some of the observer gains:

$$L_1 = 10^2 \times \begin{bmatrix} -1668.3 \\ 11.3 \\ -136.5 \\ 14.6 \\ -94.6 \end{bmatrix}, \; L_8 = 10^2 \times \begin{bmatrix} -1668.3 \\ 11.3 \\ -136.4 \\ 14.7 \\ -94.6 \end{bmatrix}, \; L_{16} = 10^2 \times \begin{bmatrix} -834.86 \\ 6.69 \\ -68.30 \\ 7.40 \\ -47.36 \end{bmatrix}$$

$$L_{32} = 10^2 \times \begin{bmatrix} -834.01 \\ 5.05 \\ -68.18 \\ 7.39 \\ -47.25 \end{bmatrix}, \; L_{64} = 10^2 \times \begin{bmatrix} -836.67 \\ 5.05 \\ -68.40 \\ 07.41 \\ -47.40 \end{bmatrix}$$

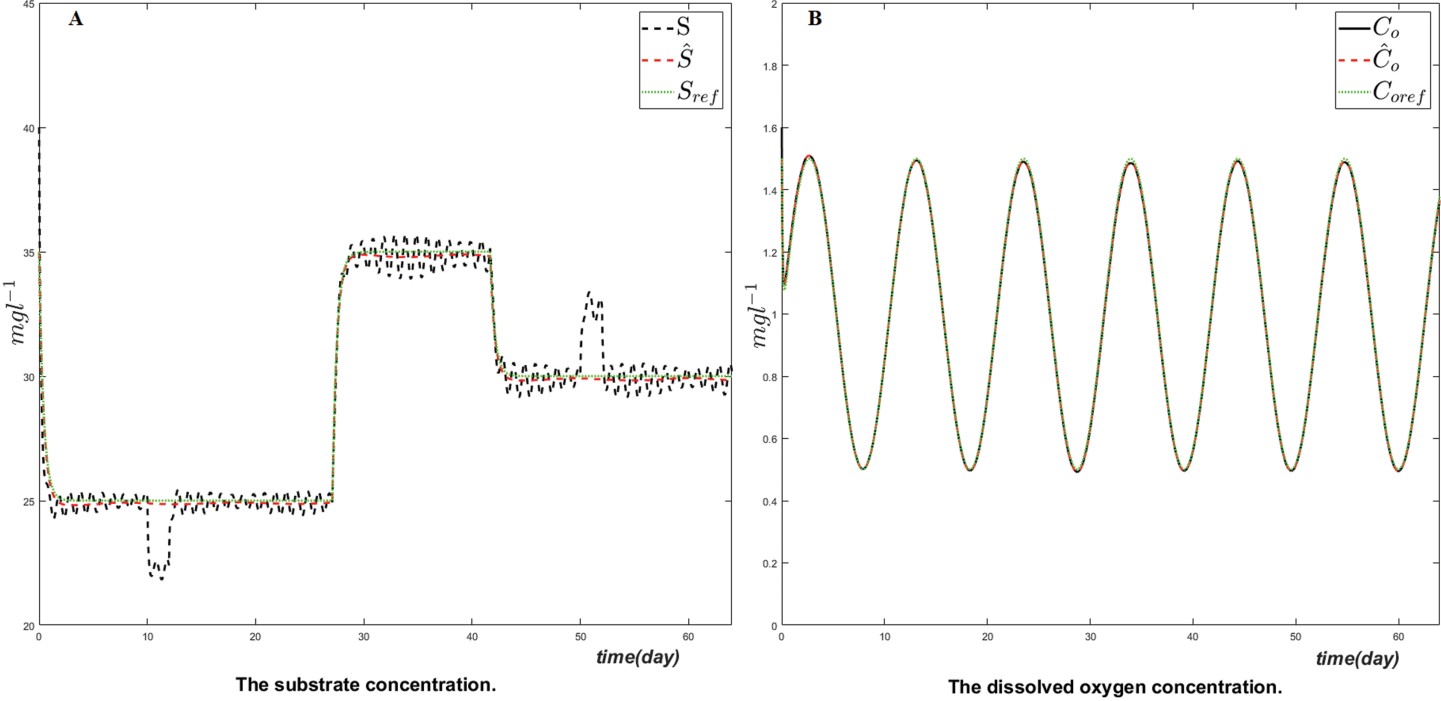

**Figure 2 The evolution of the controlled variables the substrate (A) and the dissolved oxygen (B) concentrations, their estimates and their corresponding reference trajectories.**

To test the robustness of the proposed method, two type of disturbances are introduced during the simulation as follows:

- A variation of sinusoidal form (with a period of one day and amplitude of 5% of the daily average value $S_{in}$ = 200 g/l) in the influent substrate;
- Two changes over a period of two days in two kinetic parameters respectively $\mu_{max}$ (10%) at $t$ = 10 days, and $Kc$ (10%) at $t$ = 50 days

The goal of the proposed control strategy is to follow the output references as closely as possible. This is well illustrated in Fig. 2 where a comparison is given between the true simulated value, the estimated and the corresponding reference trajectory respectively for the dissolved oxygen concentration $C_o$ and the substrate $S$. The results show the ability of the obtained control law to track the reference trajectories of the controlled variables after a short transient response despite the changes of the set-points. The perturbations of substrate and dissolved oxygen regulation due to considered disturbances are favorably rejected by the controller especially for the dissolved oxygen.

Figure 3 shows the manipulated variables respectively the dilution rate $D$ and the air flow rate $Kla$. Their dynamics change whenever the reference model changes and the control objective is clearly reached.

Figure 4 represents respectively the reconstructed biomass $\hat{X}$ and recycled biomass $\hat{X}_r$. As indicated previously, these two variables are not considered in the tracking problem. Nevertheless, it can be seen in these results that the estimated values of theses states

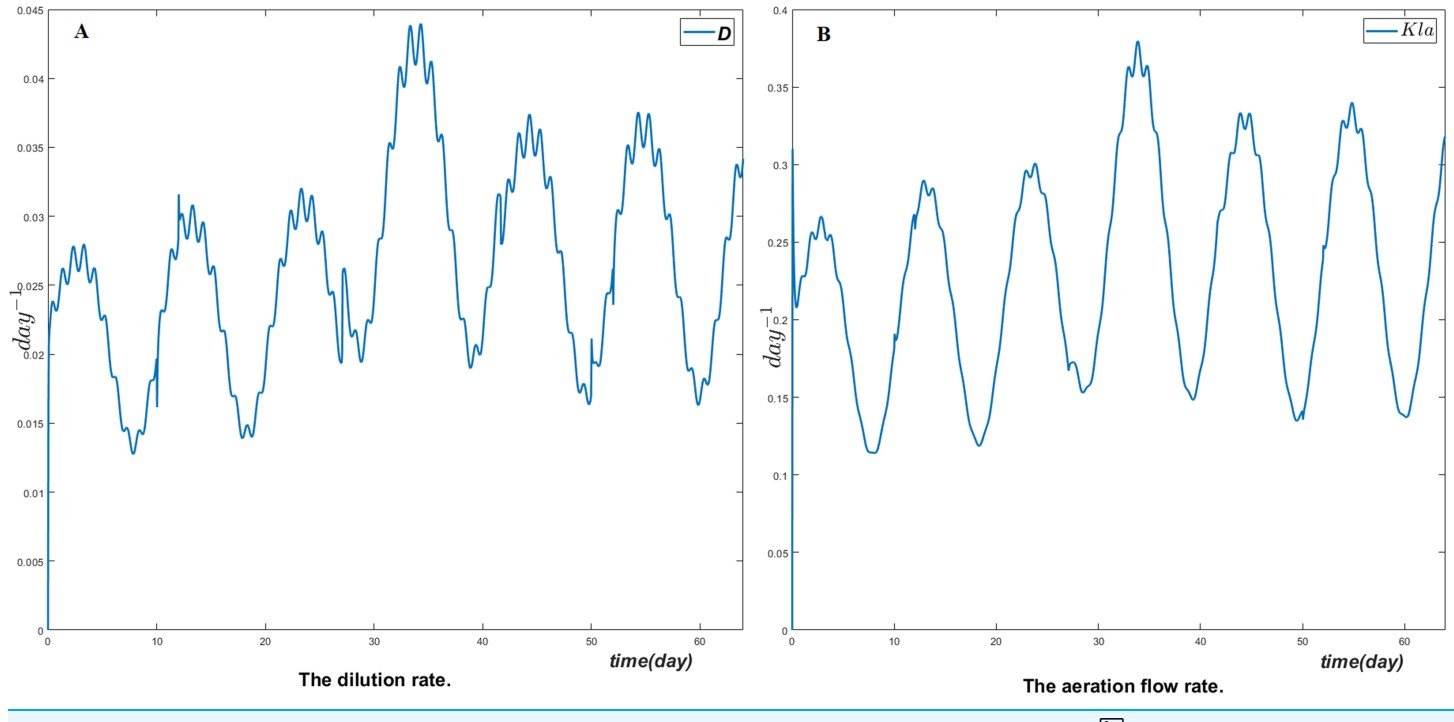

**Figure 3** The manipulated variables (A) the dilution rate and (B) the aeration flow rate.   

**Figure 4** Comparison between (A) the biomass and (B) the recycled biomass with their estimates.

variables are in excellent agreement with their corresponding true simulated values. Despite perturbation in the influent substrate and changes in kinetic parameters, the estimates are smooth and reliable.

## CONCLUSION

In this article, an observer based robust fuzzy tracking controller has been first developed then applied to a strongly nonlinear process with a biological nature. The nonlinear system is equivalently represented by a TS multi-model. Considering that the states are not fully accessible, the stability analysis and design of TS fuzzy system via an observer-based tracking controller satisfying the $H_\infty$ performance requirement has been investigated. Controller and observer gains are obtained by solving a set of LMIs. These theoretical points has been then applied to an activated sludge process where the only measured variable available online is the concentration of dissolved oxygen, which is the most realistic and economical choice. The tracking problem of reference trajectories of two state variables (the dissolved oxygen and the substrate) making use of two manipulated variables (the dilution rate and the aeration flow rate) has been addressed. The numerical simulation results illustrate the effectiveness of the proposed method and show good state estimation and tracking performances. Extension of the proposed approach to fault tolerant control is the focus of our future work, we investigate a TS fuzzy Proportional Integral observer-based fault-tolerant control subject to actuators and sensors fault of TS fuzzy positive systems with saturation on the inputs.

## PROOF OF THEOREM 2

Let consider the following Lyapunov function:

$$V(\bar{e}_o) = \bar{e}_o(t)^T P_o \bar{e}_o(t) \tag{47}$$

To achieve the performance (41) and ensure the stability of system (40), the following condition must be realized:

$$\dot{V}(\bar{e}_o)) + \bar{e}_o^T(t)R\bar{e}_o(t) - v^2\bar{\omega}(t)^T\bar{\omega}(t) < 0 \tag{48}$$

Let consider the derivative of the Lyapunov function $V(\bar{e}_o)$:

$$\dot{V}(\bar{e}_o) = \bar{e}_o^T(t)((\bar{A}_i - \bar{L}_i\bar{C})^T P_o + P_o(\bar{A}_i - \bar{L}_i\bar{C}))\bar{e}_o(t) \\ + \bar{\omega}(t)^T P_o \bar{e}_o(t) + \bar{e}_o^T(t) P_o \bar{\omega}(t) \tag{49}$$

Using Lemma 1 leads to:

$$\bar{\omega}(t)^T P_o \bar{e}_o(t) + \bar{e}_o^T(t) P_o \bar{\omega}(t) \le \eta\bar{\omega}(t)^T\bar{\omega}(t) + \eta^{-1}\bar{e}_o^T(t) P_o P_o \bar{e}_o(t) \tag{50}$$

(49) and (50) leads to:

$$\dot{V}(\bar{e}_o) + \bar{e}_o^T(t)R\bar{e}_o(t) - v^2\bar{\omega}(t)^T\bar{\omega}(t) \le \sum_{i=1}^{n_r} h_i(\hat{z})[\bar{e}_o^T(t)((\bar{A}_i - \bar{L}_i\bar{C})^T P_o \\ + P_o(\bar{A}_i - \bar{L}_i\bar{C}) + R + \eta^{-1}P_o P_o)\bar{e}_o(t) \\ + (\eta - v^2)\bar{\omega}(t)^T\bar{\omega}(t) \tag{51}$$

Consequently (48) will be achieved if the following condition (51) holds for $i = 1, \ldots n_r$:

$$(\bar{A}_i - \bar{L}_i\bar{C})^T P_o + P_o(\bar{A}_i - \bar{L}_i\bar{C}) + R + \eta^{-1}P_oP_o < 0 \tag{52}$$

$$\eta - v^2 < 0 \tag{53}$$

Applying Schur complement to (52) we get:

$$\begin{bmatrix} P_o(\bar{A}_i - \bar{L}_i\bar{C}) + (\bar{A}_i - \bar{L}_i\bar{C})^T P_o + R & P_o \\ P_o & -\eta I \end{bmatrix} < 0 \tag{54}$$

By using the variable change $Z_i = P_oL_i$, the BMIs (bilinear matrix inequalities) (53) are transformed into the LMIs given by (42). This achieves the proof of Theorem 1.

### Funding
The authors received no funding for this work.

### Competing Interests
The authors declare that they have no competing interests.

### Author Contributions
- Abdelmounaim Khallouq conceived and designed the experiments, performed the experiments, analyzed the data, performed the computation work, prepared figures and/or tables, and approved the final draft.
- Asma Karama performed the experiments, analyzed the data, authored or reviewed drafts of the paper, and approved the final draft.
- Mohamed Abyad analyzed the data, performed the computation work, authored or reviewed drafts of the paper, and approved the final draft.

### Data Availability
    Code are available in the Supplemental Files.

### Supplemental Information
Supplemental information for this article can be found online at http://dx.doi.org/10.7717/peerj-cs.458#supplemental-information.

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
