# Peer review of "Observer based robust H∞ fuzzy tracking control: application to an activated sludge process"

_PeerJ Computer Science, doi:10.7717/peerj-cs.458_

## Round 0.1 · original submission · Major Revisions

The manuscript investigates an interesting topic. However, a major improvement is necessary as the basic English writing has to be smoothed by a proofread. The validation of the presented method should be given with more details, where a comparison study is essential to demonstrate the performance. Please following the reviewers' comments carefully and address all these comments in the revised version.

Reviewer 1 ·

Basic reporting

The paper use a clear English, has a strong technical background, a relevant but not complete literature review, with appropriate paper structure and figures, as well as clear definitions of all terms and theorems, and detailed proofs.

The paper should be considered for publication, after solving some elements/issues detailed in the following review.
1. The most important issue is the lack in explaining the novelty of your proposed method.
2. The next most important item is the positioning of your paper in the recent international context.
3. Some minor mathematical gaps were detected.
4. The least important points are related to the English.

Experimental design

The paper is according to the aims and scope of the journal.
The research question is quite well defined, but needs some further investigation.
The research complies with high technical and ethical standards.
The method is sufficiently described from the reproduction aspect.

Validity of the findings

Impact and novelty not sufficiently assessed, but existing. Some refinement are recommended in the review.
Details on the data used are not explicitly given.
Conclusion is appropriately stated.

Additional comments

Paper Review
As a summary of this review:
The paper uses a clear English, has a strong technical background, a relevant but not complete literature review, with appropriate paper structure and figures, as well as clear definitions of all terms and theorems, and detailed proofs.
The paper should be considered for publication, after solving some elements/issues detailed in the following :
1. Your most important issue is the lack in explaining the novelty of your proposed method.
2. The next most important item is the positioning of your paper in the recent international context.
3. Some minor mathematical gaps were detected.
4. The least important points are related to the English.
please find details in the file attached.

Annotated reviews are not available for download in order to protect the identity of reviewers who chose to remain anonymous.

Reviewer 2 ·

Basic reporting

The paper is generally well written.
The authors should improve the list of references with papers like: https://doi.org/10.1016/j.sigpro.2014.09.011
This is needed especially since the problem of using TS fuzzy approach is not new in the case of the wastewater treatment processes.
The paper needs a serious proof reading. There are a lot of typos such as: Takagi Seguno fuzzy model.
The authors should also clearly present which parts are new and which parts are taken from the literature. For example: Lemmas are new or not because there is no citation attached to them.

Experimental design

The authors should test the robust solution in the case of some uncertainties. I recommend to consider some variation of the maximum specific growth rate or other parameters from the model. This is very important since this type of process are highly affected by uncertainties.

Validity of the findings

From my own works I didn't manage to have a observer for this model using only the dissolved oxygen concentration as measured variable (it would interesting to linearize the model in different operating points and test the observability of the linear model. I did that and the results are not promising). Maybe this method can offer this possibility.

Additional comments

In my opinion the work can be improved taken into consideration the above comments.

---

## Round 0.2 · Minor Revisions

As the revised version version, the quality of the submission has been improved with publishable contents. However, reviewer still has some concerns regarding the literature review. Basically, the exitinsg estimation methods should be covered, for instance, Kalman, Hinf, Luenberger, etc. In particular, if the process is affected by non-Gaussian measurements, the minimum entropy filtering should also be briefly introduced to enrich the motivation and background of the research contributions.

Reviewer 2 ·

Basic reporting

The authors provided an improved version of the manuscript. I still think that some other papers that were dealing with estimation in the case of wastewater treatment processes (using Kalman, Hinf, Luenberger etc.) should be included in the references section.

Experimental design

OK

Validity of the findings

OK

Additional comments

OK

---

## Round 0.3 · accepted · Accept

After the major and minor revision, the quality of the paper has been enhanced and the contributions have been highlighted in terms of H_inf robust fuzzy tracking control. Therefore, I recommend to accept the current version.

Reviewer 2 ·

Basic reporting

OK

Experimental design

OK

Validity of the findings

OK

Additional comments

OK